# Pharmacopuncture Effects on Insomnia Disorder: Protocol for a Multi-Site, Randomized, Acupuncture-Controlled, Clinical Trial

**DOI:** 10.3390/ijerph192416688

**Published:** 2022-12-12

**Authors:** Jung-Hwa Lim, Jae-Hyok Lee, Chan-Young Kwon, Sang-Hyup Lee, Chang-Wan Kang, Eun Cho, Hyun-Woo Kim, Jun-Hee Cho, Bo-Kyung Kim

**Affiliations:** 1Department of Neuropsychiatry, School of Korean Medicine, Pusan National University, 49, Busandaehak-ro, Yangsan-si 50612, Republic of Korea; 2Pusan National University Korean Medicine Hospital, 20, Geumo-ro, Yangsan-si 50612, Republic of Korea; 3Department of Neuropsychiatry, College of Korean Medicine, Semyung University, 63, Sangbang 4-gil, Chungju-si 27429, Republic of Korea; 4Department of Oriental Neuropsychiatry, College of Korean Medicine, Dong-Eui University, 52-57, Yangjeong-ro, Busan-si 47227, Republic of Korea; 5Department of Korean Medical Classics, College of Korean Medicine, Dong-Eui University, 52-57, Yangjeong-ro, Busan-si 47227, Republic of Korea; 6Industrial Management, Big Data Engineering Major, Dong-Eui University, 176, Eomgwang-ro, Busan-si 47340, Republic of Korea; 7College of Pharmacy, Sookmyung Women’s University, 100, Cheongpa-ro 47-gil, Seoul-si 04310, Republic of Korea; 8Department of Neurology, Pusan National University Yangsan Hospital, 49, Busandaehak-ro, Yangsan-si 50612, Republic of Korea

**Keywords:** pharmacopuncture, acupuncture, insomnia disorder, pragmatic clinical trial

## Abstract

Insomnia is a common health problem that can lead to various diseases and negatively impact quality of life. Pharmacopuncture is a new type of acupuncture that involves applying herbal medicine extracts to acupoints. Korean medicine doctors frequently use it to treat insomnia disorder. However, there is insufficient evidence to support the effectiveness and safety of pharmacopuncture for insomnia disorder. We designed a pragmatic randomized controlled trial to compare the effectiveness of pharmacopuncture and acupuncture for insomnia disorder. This multi-site, randomized, acupuncture-controlled trial will enroll 138 insomnia patients. The subjects will be randomly assigned to one of two groups, pharmacopuncture or acupuncture, at a 2:1 ratio. For 4 weeks, the participants will receive ten sessions of pharmacopuncture or acupuncture treatment and will be followed up for 4 weeks after the treatment ends. The Pittsburgh Sleep Quality Index score is the primary outcome measure. Insomnia severity index score, sleep parameters recorded using actigraphy and sleep diaries, physical symptoms associated with insomnia, emotions, quality of life, medical costs, and safety are the secondary outcome measures. The findings of this trial willprovide evidence that will be useful in clinical decision-making for insomnia treatment strategies.

## 1. Introduction

Insomnia disorder is defined according to the Diagnostic and Statistical Manual of Mental Disorders, fifth edition (DSM-5) diagnostic criteria as difficulty falling asleep, maintaining sleep, or waking up too early in the morning despite adequate sleep opportunities, resulting in impaired daytime function [1]. Insomnia disorder is highly prevalent; various studies conducted worldwide have revealed that insomnia affects 10–30% of the population, with some estimates as high as 50–60% [2,3,4,5]. According to an epidemiologic study conducted in South Korea, approximately 20% of adults have insomnia [6]. Insomnia is linked to reduced productivity, poor job, or academic performance, an increased risk of workplace or traffic accidents, and an increased vulnerability to a variety of medical conditions such as psychiatric disorders, cognitive impairment, cardiovascular diseases, and metabolic diseases [7,8,9,10,11]. As a result, insomnia can negatively impact one’s quality of life as well as be a significant financial burden [12,13].

Socioeconomic status (SES)—defined by income, education, and employment—influences health disparities and social inequities and can affect public health outcome by different processes [14]. Recently, several studies reported that sleep health and sleep disturbance are correlated to SES. Poor income, low education, unemployment, and difficult living conditions are reportedly associated with poor sleep quality, insomnia, and excessive daytime sleepiness [15,16]. Moreover, a meta-analysis showed that the influence of SES on sleep health can be measured objectively and quantitatively through actigraphy or polysomnography [17].

In Western medicine, pharmacological treatment and cognitive behavioral therapy (CBT) are used to treat insomnia [18,19,20]. Pharmacotherapy, in particular, benzodiazepines are effective for the short-term (3–4 weeks) treatment of insomnia [20]. However, clinical guidelines state that short-term hypnotics administration should be supplemented with behavior and cognitive therapies when possible [4]. Moreover, long-term use of benzodiazepine increases the risk of abuse, tolerance, dependence, and associated medical complications such as falls, fractures, and impaired attention, psychomotor function, and cognitive function especially in older adults [21,22,23,24]. Meanwhile, CBT for insomnia (CBT-I) is recommended as the initial treatment modality for patients with chronic insomnia, with sleeping pills considered an adjunct to CBT-I [18,19]. CBT-I is not only effective in improving insomnia [25,26] but also comorbid insomnia with sleep apnea, arthritis, and coronary artery disorder [27,28]. However, some patients fail to achieve any benefit from this treatment [29], and there are some challenges in implementing CBT-I, including the lack of trained therapists, low patient motivation, and high burden of cost and time [29]. Recently, Internet-based CBT-I is shown to be as effective as in-person practitioner-administered CBT-I, and there are debates on the extent of therapist’s involvement in this treatment [30].

Acupuncture is the most frequently used treatment approach for insomnia in Korean medicine, as reported by a 2016 survey [31] and is reimbursed by the National Health Insurance (NHI) [32]. A systematic review including 15 randomized controlled trials (RCTs) with 1108 patients reported that acupuncture was more effective than sham or placebo acupuncture in improving subjective sleep quality, insomnia severity, and various sleep components [33]. The Clinical Practice Guideline of Korean Medicine for Insomnia Disorder states that acupuncture, as monotherapy and in combination with sleeping pills, should be considered in treating insomnia. The Clinical Practice Guideline of Korean Medicine for Insomnia Disorder was the first of its kind in South Korea to recommend traditional Korean medical interventions for treating insomnia. However, it was limited in that it did not recommend CBT-I as an initial treatment [34].

Pharmacopuncture is a new type of acupuncture therapy that involves injecting herbal medicine extracts into acupoints, thus producing a combination of mechanical and chemical effects. Its benefits include a relatively quick effect and ease of administered dosage control. Additionally, since the herbal medicine is administered and absorbed directly into the acupoints without direct involvement of the gastrointestinal tract, this treatment can be extended to patients who cannot consume herbal medicine orally [35]. Pharmacopuncture is frequently used for the treatment of insomnia, and more than one-third of Korean medicine doctors reported using this treatment for insomnia in 2016 [31]. A 2020 survey showed that the most commonly used type of pharmacopuncture for insomnia is placental pharmacopuncture, followed by Hwangryunhaedok-tang, Ouhyul, and Bufonis venenum pharmacopuncture [36]. A systematic review reported that the decrease in PSQI score after pharmacopuncture therapy was significantly greater than that after pharmacotherapy or acupuncture [37]. Korean medicine doctors theorize that pharmacopuncture may effectively and safely treat insomnia. However, the implementation of this therapy is limited in clinical settings because of insufficient clinical evidence and the high cost of procedure, which is due its exclusion from national health insurance coverage [36].

However, RCTs comparing the effectiveness and safety of pharmacopuncture and acupuncture on insomnia disorder are scarce. Furthermore, to the best of our knowledge, there have been no RCTs conducted in Korea comparing the effectiveness and safety of pharmacopuncture and acupuncture on the improvement of insomnia. 

Therefore, a well-designed RCT that can reflect the clinical status and context of Korean medicine is needed. We will perform a multi-site, pragmatic, randomized, acupuncture-controlled trial to investigate the effectiveness and safety of pharmacopuncture compared with acupuncture covered by NHI for insomnia disorder. The results of this study will provide clinical and economic evidence that will be helpful in clinical decision-making for various insomnia treatment strategies used in Korean medical clinical settings.

## 2. Materials and Methods

### 2.1. Study Design

This multi-site, pragmatic RCT was conducted at Pusan National University Korean Medicine Hospital, Dong-eui University Korean Medicine Hospital, and Semyung University Korean Medicine Hospital in Chungju. The study protocol was approved by each research institution’s institutional review board (IRB) before trial initiation (Pusan National University Korean Medicine Hospital, Institutional Review Board: PNUKHIRB-2021004; Semyung University Korean Medicine Hospital in Chungju, Institutional Review Board: SMCJH 2110-12; and Dong-eui University Korean Medicine Hospital, Institutional Review Board: DH-2021-13). The IRB will approve any changes to the study protocol, including recruitment advertisements, informed consent forms, and case report forms (CRF). The study protocol was registered with the Clinical Research Information Service (CRIS) (no. KCT0006803).

### 2.2. Participants Recruitment and Study Schedule

A total of 138 participants with insomnia will be recruited and registered for competitive enrollment. Potential participants will be recruited through internal advertisements on bulletin boards and the homepages of each clinical institution, external advertisements in local newspapers, subway advertisements, and flyer advertisements. The investigators, who are qualified neuropsychiatrists of Korean medicine, will explain the purpose, treatments, procedures of the clinical trial, and any risks and potential for adverse effects that may occur during this trial procedure. Participants will be asked to sign informed consent forms after investigators sufficiently explain this clinical trial, and receive a copy of the consent form. The investigators will then diagnose the insomnia disorder according to DSM-5 criteria. Eligible participants will be randomly assigned to the pharmacopuncture or acupuncture group in a 2:1 ratio at visit 1. They will visit the research institution 12 times and undergo 10 treatments 5 times every 2 weeks for 4 weeks (visits 1–10). Within 1 week of treatment completion, a post-intervention assessment will be conducted (visit 11), and a follow-up assessment (visit 12) will be performed 4 weeks after treatment completion. The timeline of enrollment, treatment, and evaluation of the participants is summarized in Table 1, and a flowchart of the trial is presented in Figure 1. Additionally, 15 consecutive participants from each group enrolled at the Pusan National University Korean Medicine Hospital after random randomly assignment will undergo polysomnography, melatonin, and cortisol saliva studies on the day before the initiation of treatment (visit 1) and at the post-intervention evaluation (visit 11).

### 2.3. Sample Size

In this trial, the ratio of the pharmacopuncture group to the acupuncture group will be 2:1 in order to obtain data on the side effects (safety) of the pharmacopuncture while securing statistical power [38,39]. The sample size of this study was determined based on the study by Xian et al. [40]. The mean differences in PSQI scores before and after treatment in the experimental and control groups were estimated to be 6.9 (standard deviation = 2.51) and 4.93 (standard deviation = 2.70), respectively. Based on this, the effect size calculated using G Power software, version 3.1.9.4 (Franz Faul, Christian-Albrechts-Universittt Kiel, Kiel, Germany) was 0.92. The number of participants required is then 74 for the experimental group and 37 for the control group, given a 2:1 allocation ratio, two-sided significance level of 0.05, and 95% statistical power. Considering a dropout rate of 20%, the total sample size of the experimental and control groups will be 92 and 46, respectively.

### 2.4. Eligibility Criteria: Inclusion Criteria

(1)Male and female participants aged 19–80 years.(2)Total Insomnia Severity Index (ISI) score ≥ 15.(3)Diagnosis of insomnia disorder according to DSM-5 criteria.(4)Voluntary participation and signing of an informed consent form after adequate explanation of the purpose and procedure of this clinical trial and any risks and potential for adverse effects from this trial procedure.

### 2.5. Eligibility Criteria: Exclusion Criteria

(1)Changes in the type or dosage of regularly used sleeping pills within the last 4 weeks(2)Korean medicine treatment (acupuncture, moxibustion, cupping, and herbal medicine) to improve insomnia within the last 4 weeks.(3)Initiation of health supplements or other non-pharmacological therapies (e.g., cognitive behavioral therapy and meditation) to improve insomnia within the last 4 weeks or plans to start the therapies during the clinical trial period.(4)Current participation in other clinical trials with interventions within the last 4 weeks(5)Shift workers.(6)Obvious pain that interferes with sleep or diseases causing insomnia.(7)Treatment for unstable (controllable) schizophrenia, bipolar disorder, and other mental disorders during the last 6 months or an anxiety or depression subscale score of ≥11 on the Hospital Anxiety and Depression Scale.(8)Current treatment for serious chronic diseases or terminal diseases (malignant tumor, tuberculosis, chronic liver disease, interstitial pneumonia, chronic renal disease, chronic heart disease, and other rare metabolic diseases).(9)Abnormal hormone levels on thyroid function tests (abnormal free T4 and thyroid stimulating hormone < 0.1 uIU/mL or >5.1 uIU/mL).(10)Clinically significant abnormalities in blood chemistry (aspartate aminotransferase and alanine aminotransferase > 2 times the upper limit; serum creatinine ≥ 1.5 times the normal upper limit).(11)Electrolyte abnormalities (5% more or less than the normal range).(12)Hypertension not controlled with antihypertensive drugs (systolic blood pressure >160 mmHg and diastolic blood pressure > 90 mmHg).(13)Diabetes mellitus not controlled with oral hypoglycemic agents or insulin-dependent diabetes (hemoglobin A1c level ≥ 7%).(14)Use of hemostatic agents (e.g., Greenmono, Advate, Monoclate-P, Facnyne, and BenFix) for cardiovascular diseases or hemostatic disorders.(15)Pregnancy, lactation, and disagreement with contraception methods (dual contraception, intrauterine contraceptive devices, and spermicides) during the clinical trial.(16)Difficulty with adherence to the study protocol.(17)Deemed unsuitable for participation by the investigator.

### 2.6. Randomization, Allocation Concealment, and Blinding

A statistical expert not involved in the assessment and intervention will generate the allocation sequence using a stratified block randomization method with the research institution as a stratification factor using the statistical program SAS 9.4 (SAS Institute, Cary, NC, USA). The randomization list will be delivered to each institution and stored in sealed opaque envelopes in a double-locked cabinet. The investigator will open the envelope to randomly assign the participants to the pharmacopuncture or acupuncture groups. The investigator will store the open envelopes in a double-locked cabinet and record the randomized number assigned to each patient on the electronic medical chart.

Due to the nature of the interventions, it will be impossible to achieve blinding of participants or health practitioners. Outcome measures will be assessed prior to treatment in separate rooms by assessors who are not involved in allocation and interventions to ensure assessor blinding. The assessors participating in this study are residents at the Department of Neuropsychiatry of Korean Medicine and clinical research coordinators who have completed research ethics education and standard operating procedure training related to this trial before trial initiation.

### 2.7. Intervention

#### 2.7.1. Experiment Group: Pharmacopuncture

Since this trial is a pragmatic study that reflects Korean medical clinical practice, in which personalized treatment is provided according to the patient’s clinical status, the types and methods of pharmacopuncture, acupoints, amount of injection, and depth of injection have not been predetermined. Rather, the health practitioners (Korean medicine doctors) will determine the treatment procedure based on the participant’s clinical status. However, in this trial, the types of pharmacopuncture are limited to placental, Hwangryunhaedok-tang, and Bufonis venenum pharmacopuncture; these are already performed at each research institution and are the most commonly used insomnia treatments in a previous survey [36].

For pharmacopuncture, the acupoints GV20, Ex-HN3, Ex-HN22, GB12, HT7, CV4, and CV6, which were used for insomnia treatment in a previous study [41], will be used, but we will allow the Korean medical doctors to change the acupoints used according to their clinical decision based on the patient’s symptoms and condition.

#### 2.7.2. Control Group: Acupuncture

The comparator treatment will be acupuncture, which is the most commonly used method for treating insomnia in Korean medical practice according to a previous survey [31]. Similar to the pharmacopuncture treatment, the health practitioners will decide the acupuncture protocol in terms of the acupuncture type, acupoints, needle type, depth of insertion, and retention time, depending on the patient’s condition.

Nevertheless, based on previous studies on insomnia and consensus between Korean medicine specialists in neuropsychiatry [42,43], we primarily selected the acupoints GV20, EX-HN3, and bilateral HT7, PC6, BL63, and KI4, but we will allow the practitioners to modify the acupoints used according to their clinical decisions. All treatment-related parameters will be recorded for analysis.

Both pharmacopuncture and acupuncture will be performed by Korean medicine doctors with more than 2 years of experience in acupuncture practice. All practitioners will have completed the standard operating procedure training related to pharmachopuncture and acupuncture administration before this trial’s initiation. To improve adherence to the intervention, treatment, and assessment will be performed through reservation, and participants will be notified of the date of their visit in advance via phone. In addition, investigators will check the participants’ concomitant drug changes and adverse events (AEs) at every visit.

### 2.8. Concomitant Treatment

Medications that have been used 4 weeks prior to participation and medications used for other medical conditions (hypertension, diabetes mellitus, hyperlipidemia, and pain) will be allowed. The participants will be allowed to receive medical care for other conditions that do not affect the treatment results during the trial period. All details regarding concomitant treatment will be investigated at every visit and recorded in the medical records and CRFs.

### 2.9. Outcome Measures

#### 2.9.1. Primary Outcome

The primary outcome measure will be the mean difference in the total Pittsburgh Sleep Quality Index (PSQI) scores pre- and post-intervention (between visits 1 and 11) between the two groups. The PSQI is the most commonly used self-report questionnaire to assess sleep quality and disturbances over the past month. It comprises 19 self-rated items and evaluates seven components: subjective sleep quality, sleep latency, sleep duration, habitual sleep efficiency, sleep disturbances, use of sleep medication, and daytime dysfunction. The total PSQI score ranges from 0 to 21, with higher scores indicating poorer sleep quality [44]. The PSQI score will be assessed at baseline, visit 6, post-intervention, and follow-up visits.

#### 2.9.2. Secondary Outcomes

The secondary outcomes will be changes in the ISI, PSQI, and visual analog scale (VAS) scores of subjective symptoms accompanying insomnia, and the difference in quantity and quality of sleep recorded using a sleep diary and actigraphy during the trial period. The ISI and VAS scores of subjective symptoms accompanying insomnia, sleep diaries, and actigraphy will be checked at the baseline, visit 6, post-intervention, and follow-up visits. In addition, changes in scores on the Core Seven-Emotions Inventory Short Form (CSEI-S), Pattern Identification Tool-Insomnia (PIT-Insomnia), and body composition test between the pre- and post-treatment periods will be analyzed. The ISI is a self-report questionnaire designed to assess a patient’s subjective perception of insomnia severity. It consists of seven items rated on a 5-point Likert scale. The total ISI score ranges from 0 to 28 [45], with a higher score indicating more severe insomnia; the cutoff point for clinically significant insomnia is 15 [46].

We will assess the improvement in physical symptoms accompanying insomnia, such as headache, dyspepsia, and fatigue, using the VAS scale.

The PIT-insomnia, developed by Lee et al. [47] is a self-report questionnaire for assessing the pattern identification of insomnia.

The CSEI-S is a self-rating questionnaire used to assess the quantity of seven basic human emotions: joy, anger, thought, depression, sorrow, fear, and fright, based on the seven emotions theory in Korean medicine [48].

A sleep diary is a useful tool for obtaining sleep-related information and monitoring the treatment progress [49]. Eligible participants will be asked to complete a sleep diary after waking up daily. The sleep diary items included bedtime, waking time, sleep onset latency, sleep duration, number of awakenings, use of medications, and sleep satisfaction scores ranging from 0 to 100, with higher scores indicating greater sleep satisfaction.

The eligible participants will be asked to place an actigraphy device (ActiGraph wGT3X-BT; MTI Health Services Company, Pensacola, FL, USA) on their non-dominant wrists to obtain objective sleep-related information. Records from the actigraphy device at baseline, visit 6, post-intervention, and follow-up visits will be downloaded and saved. The extracted sleep-related variables will be used to compare the changes between the groups and their consistency with sleep diary records.

We will calculate the total time in bed from the sleeping time and waking time recorded in the sleep diary. The sleep diary assesses the sleep onset latency and total sleep time. We will also calculate sleep efficiency from the items entered in the sleep diary. Additionally, we will analyze objective sleep-related data such as sleep onset latency, total sleep time, and sleep efficiency based on actigraphy findings.

Body composition of the participants will be measured using the InBody S-10 (Biospace, Seoul, Republic of Korea) at baseline and post-intervention visits. We will compare the changes in body weight, skeletal muscle mass, body fat mass, body mass index, visceral fat area, and extracellular space between the two groups.

#### 2.9.3. Quality of Life Assessment

Quality of life will be measured at the baseline, post-intervention, and follow-up visits using the 5-level EuroQoL-5 dimension [50], the EuroQol-visual analog scale [51], the 36-Item Short Form Health Survey [52], and The Korean Health-Related Quality of Life Instrument with eight items [53].

#### 2.9.4. Cost Data

In this study, direct medical and non-medical costs and costs of productivity loss will be considered. Direct medical costs include expenses associated with pharmacopuncture, prescribed hypnotics, first or revisit diagnoses, and examination costs. Direct non-medical cost consists of the round-trip expenses for participating in clinical trials. The productivity loss costs will be calculated by multiplying the lost productivity scores by the hourly wage. The Work Productivity and Activity Impairment (WPAI) questionnaire will be used to calculate the lost productivity scores [54]. 

The result of the cost-effectiveness analysis will be expressed as the incremental cost-effectiveness ratio (ICER). This economic evaluation will include all types of costs and consider the quality-adjusted life-years (QALY) for effectiveness. QALY will be calculated using the EQ-5D-5L, and gross domestic product (GDP) per capita will be used as the threshold value for willingness to pay.

#### 2.9.5. Additional Evaluation

To explore the effects of the interventions on sleep parameters, including sleep architecture and sleep-related hormones, 15 participants per group enrolled at Pusan National University Korean Medicine Hospital, where polysomnography referral is available, will undergo polysomnography and saliva study of melatonin and cortisol before the start and after termination of the treatment. We will assess sleep architecture (total sleep time, sleep efficiency, sleep latency, proportions [%] of non-rapid eye movement sleep [N1, N2, N3], rapid eye movement, and arousal index), respiratory events (apnea-hypopnea index, respiratory disturbance index, respiratory arousal index), periodic limb movement index, periodic limb arousal movement, and parasomnia events using polysomnography.

#### 2.9.6. Safety Assessment

To assess safety, laboratory and electrocardiography tests will be conducted during the screening and post-intervention visits.

Investigators interviewed and recorded all AEs based on the symptoms reported by the participants at every visit during the trial period. AE severity will be graded using Spilker’s three-level criteria: mid, moderate, and severe. The causality of AEs will be rated as definitely related, probably related, possibly related, probably unrelated, definitely unrelated, or unknown based on the World Health Organization-Upsala Monitoring Centre causality assessment system.

All reported AEs and laboratory abnormalities during the trial period will be collected, and the frequency and percentage of participants who identified AEs or laboratory abnormalities in both groups will be calculated. The incidence of AEs will be compared between the two groups.

### 2.10. Data Collection and Management

To promote retention and complete follow-up, we will be in regular contact with the participants. The participants will be able to contact the researchers and obtain the information they need at any time, as contact information will be provided. Trained clinical research coordinators will record the data collected in the CRF. We will utilize electronic CRF (e-CRF) using the Internet-based clinical trial management public platform provided by the National Institute for Korean Medicine Development. To ensure the accuracy and quality of this clinical trial, all members of the research team will be educated on the standard operation procedure, including writing the CRF, data entry, and study procedures. For data quality management, the data value range for each measurement item will be set in advance. If the entered data values are outside the preset range, data queries will be issued. Only the investigator in charge of the data management will be able to access the data entered into the e-CRF. After study completion, the data will be stored for 3 years as per the relevant laws.

### 2.11. Data Monitoring and Auditing

Data monitoring will be conducted to protect the participants’ rights, ensure adherence to the study protocol, and ensure the completeness of the collected data. The Clinical Research Center of the Korea Institute of Oriental Medicine, which has no competing interests of this trial, is responsible for the data monitoring. Monitoring will be conducted by examining the consistency of the CRF and the source documents and by reviewing the safety data of the participants, including laboratory tests and AEs. Routine monitoring visits at each site will be scheduled, and additional visits will be conducted, depending on the recruitment status. If unexpected and serious AEs are associated with the intervention in this study, the Clinical Research Center of the Korea Institute of Oriental Medicine will have access to the interim results to advise on trial discontinuation. The principal investigator will make a final decision to terminate the trial.

Auditing will be performed periodically at each research institution by the Clinical Research Center of the Korea Institute of Oriental Medicine, independent of this study.

### 2.12. Criteria for Discontinuation and Withdrawal

Treatment will be discontinued and participation will be withdrawn according to the following criteria: (1) Request for discontinuation or withdrawal of consent by participants or their legal representative; (2) diagnosis of any medical condition that makes the participant ineligible for this trial; (3) incidence of any serious AEs; (4) violation of the study protocol; (5) changes in dosage or type of sleeping pills already used during the trial period (from screening test to post intervention); (6) loss to follow-up; (7) use of any medications or undergoing therapies that will affect the study results without the permission of investigators; (8) more than three consecutive absences from intervention or participation in less than seven sessions of the intervention; and (9) other conditions wherein participation is judged to be inappropriate by the investigator in charge.

### 2.13. Statistical Analysis

Continuous variables will be presented as means and standard deviations, while categorical variables will be presented as frequencies (%) and percentiles. The homogeneity of the demographic characteristics and baseline variables between the two groups will be evaluated to choose the appropriate analysis methods. The independent t-test or Mann–Whitney U-test will be used for analysis, depending on the normality of data distribution. If there were significant between-group differences in baseline variables or demographic characteristics, regression analysis with covariates (analysis of covariance) will be conducted. The primary and secondary outcome measures will be analyzed in the same manner. We will also examine the consistency between objective and subjective sleep-related data obtained from actigraphy and patient-reported questionnaires.

Additionally, we will analyze the changes in sleep parameters assessed using polysomnography and sleep-related hormones before and after treatment in participants who underwent polysomnography. Both the full analysis set (FAS) and per-protocol analyses will be conducted. The FAS is the primary analysis of effectiveness and safety. It will include all participants initially assigned to each group, except those who did not meet the eligibility criteria, those who had not received any intervention, or those who had not completed any outcome measurements. The per-protocol analysis will include participants who have completed more than seven sessions. All statistical analyses will be performed by an independent statistical expert blinded to the allocation and procedure using SPSS version 25 (IBM, Chicago, IL, USA). Statistical significance is set at *p* < 0.05.

### 2.14. Confidentiality

The personal information of the trial participants will be managed under the supervision of IRB. To ensure confidentiality, all the data collected from the trial will be treated anonymously. Third parties such as the monitoring committee for research purposes will only have access to de-identified data.

### 2.15. Dissemination Policy

The results of this clinical study will be presented at a congress and will be submitted for publication in relevant journals.

### 2.16. Trial Status

This is the latest protocol (version 1.1) approved by the IRB. This is an ongoing trial. The first participant was enrolled on 17 December 2021, and recruitment is expected to end in late 2023.

## 3. Discussion

In addition to acupuncture, which is covered by the NHI, pharmacopuncture is a common treatment modality for insomnia disorder in Korean medicine [31,32]. Pharmacopuncture is expected to have a combined beneficial effect in treating insomnia by physical stimulation through acupuncture and pharmacological treatment through herbal medicinal extracts [55]. A recent systematic review reported that pharmacopuncture can achieve significantly better sleep quality than pharmacotherapy or acupuncture [37]. However, due to methodological flaws and small sample sizes, it is difficult to draw concrete conclusions about the benefits of pharmacopuncture for insomnia and provide evidence for clinical decision-making. Another barrier to active pharmacopuncture implementation is the economic burden due to non-reimbursement of pharmacopuncture [36]. To reflect the clinical context of Korean medicine and provide evidence for clinical and policy decision-making, a pragmatic clinical study is required. Unlike traditional RCTs, which seek approval for a new drug or medical device, most Korean medical clinical studies seek to validate the efficacy of already clinically popular treatments or to compare their efficacy to that of alternative treatment strategies. As a result, pragmatic studies are appropriate for clinical trials in Korean medicine. Therefore, we designed a pragmatic RCT to compare the effectiveness of pharmacopuncture compared with acupuncture.

To the best of our knowledge, this study is the first pragmatic RCT to examine the effectiveness of pharmacopuncture in insomnia patients compared with acupuncture in south Korea. Furthermore, objective actigraphy data will be compared to subjective data from self-reported questionnaires to explore the consistency between objective and subjective sleep-related data. The findings should provide objective evidence of the effectiveness of pharmacopuncture and acupuncture for insomnia treatment. In addition, preliminary data on the effects of pharmacopuncture and acupuncture on sleep architecture and changes in sleep-related hormone levels will be provided.

In addition, we will be able to explore the influence of SES on sleep health of participants in this trial and identify the health disparities using the collected actigraphy data. We will also be able to investigate the effects of social and health inequality on the effectiveness of pharmacopuncture and acupuncture for insomnia. These findings might highlight that sleep health is an important component of public health and could provide evidence that pharmacopuncture can contribute to the public health by improving insomnia.

However, this study has some limitations. First, it is not possible to examine which type or acupoint in pharmacopuncture has the most significant effect on insomnia improvement, as we will allow for different types or methods of pharmacopuncture according to the practitioner’s clinical decision, depending on the patient’s condition. However, this policy can better reflect real-world Korean medical clinical practice, in which individualized treatment is performed. A retrospective analysis of medical records may provide complementary information regarding treatment details. We will detail the effects of these variations in the analysis and report the effectiveness of pharmacopuncture for insomnia. Second, we will only recruit patients with insomnia disorder; therefore, the generalizability of the results to insomnia patients with other comorbidities, including depression, anxiety disorders, alcohol dependence, and other disorders, might be limited. Third, the 4-week follow-up period in this trial is not sufficient to evaluate the persisting therapeutic effect of pharmcopuncture. Future studies with a follow-up period of 3 or 6 months are needed to investigate the long-term therapeutic effects of pharmcopuncture or acupuncture. Fourth, both participants and practitioners cannot be blinded to treatments because a sham acupuncture group could not be designed in this study. Therefore, we need to consider factors affecting treatment, such as participants’ expectations from pharmacopunture treatment, when interpreting the results of the study. Since the purpose of this trial is to investigate the effectiveness and safety of pharmacopuncture of uninsured treatment and acupuncture covered by NHI, this trial will not include a comparison between the intervention group and a third group treated with placebo or standard treatment as per Western medicine such as CBT-I. Further 3-arm trials including a placebo group to investigate the efficacy of pharmcopuncture compared to that of standard conventional treatment for insomnia. Finally, the insomnia-specific objective assessment tools will not be used as a primary outcome in this study. Although PSQI—the primary outcome variable in this trial—is not an insomnia-specific measurement, it allows convenient evaluation of sleep onset latency, total sleep time, and sleep efficiency, which are important parameters for assessing sleep improvement. In addition, we use ISI to examine the subjective perception of insomnia severity and actigraphy to objectively evaluate sleep. Thus, we expect to obtain subjective and objective data on the sleep improvement effects of pharmacopuncture and acupuncture.

## 4. Conclusions

Pharmacopuncture is expected to be effective in improving insomnia due to the combined effects of acupuncture and herbal medicine. The clinical evidence for the additional benefits of pharmacopuncture over acupuncture in insomnia treatment can promote further clinical research of various types and methods of pharmacopuncture. The results of this study are expected to provide evidence regarding pharmacopuncture for insomnia that will facilitate clinical decision-making and healthcare policy-making.

## Figures and Tables

**Figure 1 ijerph-19-16688-f001:**
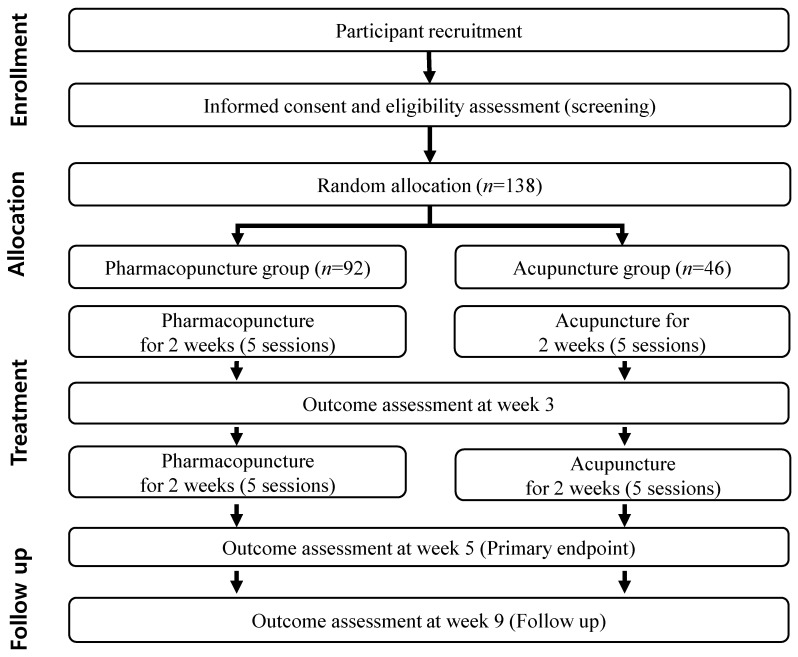
Flow diagram of the study.

**Table 1 ijerph-19-16688-t001:** Schedule of enrollment, intervention, and assessment.

Time Point	Screening	Enrollment Allocation (V1) Treatment: Total 10 Sessions (V1–10)				Post-Treatment Evaluation	Follow-Up
	Week −1	Week 0	Weeks 1–4			Week 5	Week 9
		V1	V2–5	V6	V7–10	V11	V12
Eligibility screening	O						
Informed consent	O						
Vital signs	O	O	O	O	O	O	O
Socioeconomic status, medical history	O						
ISI, HADS	O						
Laboratory tests and electrocardiogram	O					O	
Randomized allocation		O					
Pharmacopuncture (experimental)		← 5 times per 2 weeks →					
Acupuncture (control)		← 5 times per 2 weeks →					
ISI	O	O		O		O	O
PSQI		O		O		O	O
PIT for insomnia		O				O	
CSEI-S		O				O	
VAS of subjective symptoms accompanying insomnia		O		O		O	O
Quality of life (EQ-5D-5L, EQ-VAS, SF-36, HINT-8), Cost data		O				O	O
Body composition		O				O	
Request for additional treatment						O	
Sleep diary and actigraphy	O	O	O	O	O	O	O
Adverse event check		O	O	O	O	O	O
PSG, melatonin, cortisol *		O				O	

* PSG will be performed a day before visits 1 and 10. Melatonin and cortisol levels will be collected at 10 p.m. a day before visits 1 and 10 and at 7 a.m. on visits 1 and 10. Only 15 participants per group enrolled at Pusan National University Korean Medicine Hospital. ISI, Insomnia Severity Index; HADS, the Hospital Anxiety and Depression Scale; PSQI, Pittsburgh Sleep Quality Index; PIT for insomnia, Pattern Identification Tool Insomnia for insomnia; CSEI-S, The Core Seven-Emotions Inventory Short Form; VAS, visual analog scale; EQ-5D-5L, 5-level EuroQoL five-dimension questionnaire; EQ-VAS, the EuroQol-visual analogue scale; SF-36, the 36-Item Short Form Health Survey questionnaire; HINT-8, Korean Health-Related Quality of Life Instrument with 8 items; PSG, polysomnography.

## Data Availability

The data presented in this study are available on request from the corresponding author.

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
