# Peer review of "Pharmacopuncture Effects on Insomnia Disorder: Protocol for a Multi-Site, Randomized, Acupuncture-Controlled, Clinical Trial"

_ijerph, 2022, doi:10.3390/ijerph192416688_

Round 1

Reviewer 1 Report

I am glad to have the opportunity to evaluate this study. It was a very interesting manuscript on a worthy subject in the corner of many areas of research, written in a good english. For sure, this manuscript contribute to current knowledge on insomnia management.

My main issue is the lake of consideration for influence of socioeconomic status of their sample. My explanations are the following:

Understanding the etiology of socioeconomic disparities in health could assist public health authorities in preventing the morbidity of socially disadvantaged sub-groups. More research is needed to clarify the impact of sleep on the social gradient in health, as a mediator or as a consequence of socioeconomic determinants. 

Even if i believe this paper is almost complete and to my opinion deserve publication, few big concerns remains:

CONCERN 1: My main concern is the fact that authors used sociodemographic indicators in a way where it should be "socioeconomic status (SES)" which is more global and include social class, employment, location, etc... Social inequities have many health effects; one of these is a potential relationship to sleep disturbances like insomnia. Socioeconomic status (SES) is an important factor that contributes to social inequities, meaning that SES is a marker of living conditions that influence health by way of different processes. This relationship is the same regardless age and gender of participants, especially in a sample assessed both with PSQI and actigraphy. So authors should clarifiy how their results with pharmacopuncture may improve insomnia treatment in the future, knowing that some sleep disorders themselves are directly affects by socioeconomic determinants like demonstrated by several recent studies. What's the benefits compare with current cures? I also wish to know why authors decide to analyze pharmacopuncture and acupuncture alone, without a third group representing the general population (without insomnia or using different solutions to treat insomnia)?

Authors can add another table showing the distribution of results according to socioeconomic indicators like education, income, marital status, etc...

I suggest authors to extend their results and to improve their discussion by building their argumentations with the following studies:

*Towards A Socioeconomic Model of Sleep Health among the Canadian Population: A Systematic Review of the Relationship between Age, Income, Employment, Education, Social Class, Socioeconomic Status and Sleep Disparities. Eur. J. Investig. Health Psychol. Educ. 2022

*Efficacy and safety of acupuncture treatment on primary insomnia: a randomized controlled trial. Sleep Med. 2017

*Socioeconomic Position and Excessive Daytime Sleepiness: A Systematic Review of Social Epidemiological Studies. Clocks Sleep. 2022

CONCERN 2: I wish to also see authors developed a little bit how their work can help improve the current or future clinical practice (or the measurement of this mediator) related to quantitative meassures like actigraphy and polysomnography, because a recent meta-analysis show that influence of socioeconomic status (and its related variables/indicators) on sleep can be measured objectively and quantitatively, and some studies have show how health disparities in general can be identified through actigraphy or polysomnography measure in social sciences and sleep research. The current manuscript is it align with these previous findings? Do it brings something new to health disparities ? Following articles will be helpful to develop your discussion:

*Influence of socioeconomic status on objective sleep measurement: A systematic review and meta-analysis of actigraphy studies. Sleep Health. 2021

*Effect of acupuncture on sleep quality and hyperarousal state in patients with primary insomnia: study protocol for a randomised controlled trial. BMJ Open. 2016

Author Response

Reviewer 1.

I am glad to have the opportunity to evaluate this study. It was a very interesting manuscript on a worthy subject in the corner of many areas of research, written in a good english. For sure, this manuscript contribute to current knowledge on insomnia management.

My main issue is the lake of consideration for influence of socioeconomic status of their sample. My explanations are the following:

Understanding the etiology of socioeconomic disparities in health could assist public health authorities in preventing the morbidity of socially disadvantaged sub-groups. More research is needed to clarify the impact of sleep on the social gradient in health, as a mediator or as a consequence of socioeconomic determinants.

Even if i believe this paper is almost complete and to my opinion deserve publication, few big concerns remains:

CONCERN 1: My main concern is the fact that authors used sociodemographic indicators in a way where it should be "socioeconomic status (SES)" which is more global and include social class, employment, location, etc... Social inequities have many health effects; one of these is a potential relationship to sleep disturbances like insomnia. Socioeconomic status (SES) is an important factor that contributes to social inequities, meaning that SES is a marker of living conditions that influence health by way of different processes. This relationship is the same regardless age and gender of participants, especially in a sample assessed both with PSQI and actigraphy. So authors should clarifiy how their results with pharmacopuncture may improve insomnia treatment in the future, knowing that some sleep disorders themselves are directly affects by socioeconomic determinants like demonstrated by several recent studies.

==> Thank you for your valuable comments. Accordingly, we have changed the sociodemographic indicators to socioeconomic status (Table 1).

In the proposed trial, we aim to investigate how factors such as employment status, job type, education level, marital status, and number of families living together differ between the intervention and control groups. At trial completion, if a significant difference in socioeconomic status exists between those who underwent pharmacopuncture for insomnia and those who did not, we will conduct further statistical analyses considering all relevant covariates. We will also consider analyzing the impact of socioeconomic status on sleep quality and sleep distance in this trial after consulting with statistical experts. In addition, we will carefully consider the impact of socioeconomic factors when interpreting the research results. We have expressed this intent in the introduction and discussion sections of the manuscript and supported our reasoning with relevant literature.

lines 53-60 & 477-482

“Socioeconomic status (SES)—defined by income, education, and employment—influences health disparities and social inequities and can affect public health outcome by different processes [14]. Recently, several studies reported that sleep health and sleep disturbance are correlated to SES. Poor income, low education, unemployment, and difficult living conditions are reportedly associated with poor sleep quality, insomnia, and excessive daytime sleepiness [15-16]. Moreover, a meta-analysis showed that the influence of SES on sleep health can be measured objectively and quantitatively through actigraphy or polysomnography [17].”

“In addition, we will be able to explore the influence of SES on sleep health of participants in this trial and identify the health disparities using the collected actigraphy data. We will also be able to investigate the effects of social and health inequality on the effectiveness of pharmacopuncture and acupuncture for insomnia. These findings might highlight that sleep health is an important component of public health and could provide evidence that pharmacopuncture can contribute to the public health by improving insomnia.”

What's the benefits compare with current cures?

==> As described in the introduction, the recommended first-line treatment for insomnia is CBT-I; however, the applicability of CBT-I is limited by several factors, such as the lack of available experts and patient motivation and the high burden of time and cost for implementation in clinical settings. In addition, pharmacotherapy may be effective for acute insomnia treatment, but evidence for the long-term effects of this treatment is lacking, and long-term use of hypnotics can cause several side effects. Therefore, studies on safer and more readily available treatment alternatives for insomnia are needed. Pharmacopuncture can produce the benefit of both acupuncture and herbal medicine simultaneously and may potentially overcome the limitations of CBT-I and hypnotics for treating insomnia.

I also wish to know why authors decide to analyze pharmacopuncture and acupuncture alone, without a third group representing the general population (without insomnia or using different solutions to treat insomnia)?

==> In South Korea, traditional Korean medicine is recognized as conventional medicine. Acupuncture is covered by the Korean National Health Insurance (NHI) and widely used for insomnia. Notably, acupuncture is recommended for treating insomnia according to the clinical practice guideline of Korean Medicine for Insomnia Disorder v2.0. Although pharmacopuncture is not covered by the national insurance, it is commonly prescribed by many traditional Korean medicine doctors; however, the utilization of this treatment is limited by its high cost due to non-reimbursement. Therefore, we designed this trial to compare the effectiveness and safety of pharmacopuncture relative to acupuncture and conduct an economic evaluation to provide clinical and economic evidence that supports insurance coverage for pharmacopuncture. In the discussion section of our protocol, we also describe the need for a future 3-arm study with the third group comprising participants who receive placebo or standard treatment of Western medicine such as CBT-I.

Authors can add another table showing the distribution of results according to socioeconomic indicators like education, income, marital status, etc...

I suggest authors to extend their results and to improve their discussion by building their argumentations with the following studies:

==> After trial completion, we will tabulate the results according to the socioeconomic indicators of participants and analyze the impact of each indicator. In addition, we will consider the impact of participants’ socioeconomic factors on the study results. We have described this intent in the introduction and discussion sections of the manuscript as follows:

lines 53-60 & 477-482

 “Socioeconomic status (SES)—defined by income, education, and employment—influences health disparities and social inequities and can affect public health outcome by different processes [14]. Recently, several studies reported that sleep health and sleep disturbance are correlated to SES. Poor income, low education, unemployment, and difficult living conditions are reportedly associated with poor sleep quality, insomnia, and excessive daytime sleepiness [15-16]. Moreover, a meta-analysis showed that the influence of SES on sleep health can be measured objectively and quantitatively through actigraphy or polysomnography [17].”

“In addition, we will be able to explore the influence of SES on sleep health of participants in this trial and identify the health disparities using the collected actigraphy data. We will also be able to investigate the effects of social and health inequality on the effectiveness of pharmacopuncture and acupuncture for insomnia. These findings might highlight that sleep health is an important component of public health and could provide evidence that pharmacopuncture can contribute to the public health by improving insomnia.”

CONCERN 2: I wish to also see authors developed a little bit how their work can help improve the current or future clinical practice (or the measurement of this mediator) related to quantitative meassures like actigraphy and polysomnography, because a recent meta-analysis show that influence of socioeconomic status (and its related variables/indicators) on sleep can be measured objectively and quantitatively, and some studies have show how health disparities in general can be identified through actigraphy or polysomnography measure in social sciences and sleep research. The current manuscript is it align with these previous findings? Do it brings something new to health disparities ? Following articles will be helpful to develop your discussion:

==> Thank you for your valuable advice, which allowed us to gain new insights and will enrich the interpretation of this study's findings. As suggested, we will identify health disparities using actigraphy data collected in this study and investigate the impact of socioeconomic indicators on sleep health. Additionally, we will compare and analyze our findings with those of previous studies including the articles suggested by you. We have added this point in the introduction and discussion sections as follows:

lines 53-60 & 477-482

 “Socioeconomic status (SES)—defined by income, education, and employment—influences health disparities and social inequities and can affect public health outcome by different processes [14]. Recently, several studies reported that sleep health and sleep disturbance are correlated to SES. Poor income, low education, unemployment, and difficult living conditions are reportedly associated with poor sleep quality, insomnia, and excessive daytime sleepiness [15-16]. Moreover, a meta-analysis showed that the influence of SES on sleep health can be measured objectively and quantitatively through actigraphy or polysomnography [17].”

“In addition, we will be able to explore the influence of SES on sleep health of participants in this trial and identify the health disparities using the collected actigraphy data. We will also be able to investigate the effects of social and health inequality on the effectiveness of pharmacopuncture and acupuncture for insomnia. These findings might highlight that sleep health is an important component of public health and could provide evidence that pharmacopuncture can contribute to the public health by improving insomnia.”

My main issue is the lake of consideration for influence of socioeconomic status of their sample. My explanations are the following:

Understanding the etiology of socioeconomic disparities in health could assist public health authorities in preventing the morbidity of socially disadvantaged sub-groups. More research is needed to clarify the impact of sleep on the social gradient in health, as a mediator or as a consequence of socioeconomic determinants.

Even if i believe this paper is almost complete and to my opinion deserve publication, few big concerns remains:

CONCERN 1: My main concern is the fact that authors used sociodemographic indicators in a way where it should be "socioeconomic status (SES)" which is more global and include social class, employment, location, etc... Social inequities have many health effects; one of these is a potential relationship to sleep disturbances like insomnia. Socioeconomic status (SES) is an important factor that contributes to social inequities, meaning that SES is a marker of living conditions that influence health by way of different processes. This relationship is the same regardless age and gender of participants, especially in a sample assessed both with PSQI and actigraphy. So authors should clarifiy how their results with pharmacopuncture may improve insomnia treatment in the future, knowing that some sleep disorders themselves are directly affects by socioeconomic determinants like demonstrated by several recent studies.

==> Thank you for your valuable comments. Accordingly, we have changed the sociodemographic indicators to socioeconomic status (Table 1).

In the proposed trial, we aim to investigate how factors such as employment status, job type, education level, marital status, and number of families living together differ between the intervention and control groups. At trial completion, if a significant difference in socioeconomic status exists between those who underwent pharmacopuncture for insomnia and those who did not, we will conduct further statistical analyses considering all relevant covariates. We will also consider analyzing the impact of socioeconomic status on sleep quality and sleep distance in this trial after consulting with statistical experts. In addition, we will carefully consider the impact of socioeconomic factors when interpreting the research results. We have expressed this intent in the introduction and discussion sections of the manuscript and supported our reasoning with relevant literature.

lines 53-60 & 477-482

Socioeconomic status (SES)defined by income, education, and employmentinfluences health disparities and social inequities and can affect public health outcome by different processes [14]. Recently, several studies reported that sleep health and sleep disturbance are correlated to SES. Poor income, low education, unemployment, and difficult living conditions are reportedly associated with poor sleep quality, insomnia, and excessive daytime sleepiness [15-16]. Moreover, a meta-analysis showed that the influence of SES on sleep health can be measured objectively and quantitatively through actigraphy or polysomnography [17].

In addition, we will be able to explore the influence of SES on sleep health of participants in this trial and identify the health disparities using the collected actigraphy data. We will also be able to investigate the effects of social and health inequality on the effectiveness of pharmacopuncture and acupuncture for insomnia. These findings might highlight that sleep health is an important component of public health and could provide evidence that pharmacopuncture can contribute to the public health by improving insomnia.

What's the benefits compare with current cures?

==> As described in the introduction, the recommended first-line treatment for insomnia is CBT-I; however, the applicability of CBT-I is limited by several factors, such as the lack of available experts and patient motivation and the high burden of time and cost for implementation in clinical settings. In addition, pharmacotherapy may be effective for acute insomnia treatment, but evidence for the long-term effects of this treatment is lacking, and long-term use of hypnotics can cause several side effects. Therefore, studies on safer and more readily available treatment alternatives for insomnia are needed. Pharmacopuncture can produce the benefit of both acupuncture and herbal medicine simultaneously and may potentially overcome the limitations of CBT-I and hypnotics for treating insomnia.

I also wish to know why authors decide to analyze pharmacopuncture and acupuncture alone, without a third group representing the general population (without insomnia or using different solutions to treat insomnia)?

==> In South Korea, traditional Korean medicine is recognized as conventional medicine. Acupuncture is covered by the Korean National Health Insurance (NHI) and widely used for insomnia. Notably, acupuncture is recommended for treating insomnia according to the clinical practice guideline of Korean Medicine for Insomnia Disorder v2.0. Although pharmacopuncture is not covered by the national insurance, it is commonly prescribed by many traditional Korean medicine doctors; however, the utilization of this treatment is limited by its high cost due to non-reimbursement. Therefore, we designed this trial to compare the effectiveness and safety of pharmacopuncture relative to acupuncture and conduct an economic evaluation to provide clinical and economic evidence that supports insurance coverage for pharmacopuncture. In the discussion section of our protocol, we also describe the need for a future 3-arm study with the third group comprising participants who receive placebo or standard treatment of Western medicine such as CBT-I.

Authors can add another table showing the distribution of results according to socioeconomic indicators like education, income, marital status, etc...

I suggest authors to extend their results and to improve their discussion by building their argumentations with the following studies:

==> After trial completion, we will tabulate the results according to the socioeconomic indicators of participants and analyze the impact of each indicator. In addition, we will consider the impact of participants socioeconomic factors on the study results. We have described this intent in the introduction and discussion sections of the manuscript as follows:

lines 53-60 & 477-482

 Socioeconomic status (SES)defined by income, education, and employmentinfluences health disparities and social inequities and can affect public health outcome by different processes [14]. Recently, several studies reported that sleep health and sleep disturbance are correlated to SES. Poor income, low education, unemployment, and difficult living conditions are reportedly associated with poor sleep quality, insomnia, and excessive daytime sleepiness [15-16]. Moreover, a meta-analysis showed that the influence of SES on sleep health can be measured objectively and quantitatively through actigraphy or polysomnography [17].

In addition, we will be able to explore the influence of SES on sleep health of participants in this trial and identify the health disparities using the collected actigraphy data. We will also be able to investigate the effects of social and health inequality on the effectiveness of pharmacopuncture and acupuncture for insomnia. These findings might highlight that sleep health is an important component of public health and could provide evidence that pharmacopuncture can contribute to the public health by improving insomnia.

*Towards A Socioeconomic Model of Sleep Health among the Canadian Population: A Systematic Review of the Relationship between Age, Income, Employment, Education, Social Class, Socioeconomic Status and Sleep Disparities. Eur. J. Investig. Health Psychol. Educ. 2022

*Efficacy and safety of acupuncture treatment on primary insomnia: a randomized controlled trial. Sleep Med. 2017

*Socioeconomic Position and Excessive Daytime Sleepiness: A Systematic Review of Social Epidemiological Studies. Clocks Sleep. 2022

CONCERN 2: I wish to also see authors developed a little bit how their work can help improve the current or future clinical practice (or the measurement of this mediator) related to quantitative meassures like actigraphy and polysomnography, because a recent meta-analysis show that influence of socioeconomic status (and its related variables/indicators) on sleep can be measured objectively and quantitatively, and some studies have show how health disparities in general can be identified through actigraphy or polysomnography measure in social sciences and sleep research. The current manuscript is it align with these previous findings? Do it brings something new to health disparities ? Following articles will be helpful to develop your discussion:

==> Thank you for your valuable advice, which allowed us to gain new insights and will enrich the interpretation of this study's findings. As suggested, we will identify health disparities using actigraphy data collected in this study and investigate the impact of socioeconomic indicators on sleep health. Additionally, we will compare and analyze our findings with those of previous studies including the articles suggested by you. We have added this point in the introduction and discussion sections as follows:

lines 53-60 & 477-482

 Socioeconomic status (SES)defined by income, education, and employmentinfluences health disparities and social inequities and can affect public health outcome by different processes [14]. Recently, several studies reported that sleep health and sleep disturbance are correlated to SES. Poor income, low education, unemployment, and difficult living conditions are reportedly associated with poor sleep quality, insomnia, and excessive daytime sleepiness [15-16]. Moreover, a meta-analysis showed that the influence of SES on sleep health can be measured objectively and quantitatively through actigraphy or polysomnography [17].

In addition, we will be able to explore the influence of SES on sleep health of participants in this trial and identify the health disparities using the collected actigraphy data. We will also be able to investigate the effects of social and health inequality on the effectiveness of pharmacopuncture and acupuncture for insomnia. These findings might highlight that sleep health is an important component of public health and could provide evidence that pharmacopuncture can contribute to the public health by improving insomnia.

*Influence of socioeconomic status on objective sleep measurement: A systematic review and meta-analysis of actigraphy studies. Sleep Health. 2021

*Effect of acupuncture on sleep quality and hyperarousal state in patients with primary insomnia: study protocol for a randomised controlled trial. BMJ Open. 2016

Reviewer 2 Report

Thank you for the opportunity to review this protocol on the use of pharmacopuncture comparative to acupuncture for the treatment of insomnia. This is an interesting concept and the protocol to conduct this RCT is well written. I do not see any major methodology flaws in the description provided by the authors.

However, as the trial is allowing the health providers to personalize the interventions and protocols (both for pharmacopuncture and acupuncture) to the participant's clinical status, I would recommend including the variations in the final analysis if possible.

Author Response

Reviewer 2.

Thank you for the opportunity to review this protocol on the use of pharmacopuncture comparative to acupuncture for the treatment of insomnia. This is an interesting concept and the protocol to conduct this RCT is well written. I do not see any major methodology flaws in the description provided by the authors.

However, as the trial is allowing the health providers to personalize the interventions and protocols (both for pharmacopuncture and acupuncture) to the participant's clinical status, I would recommend including the variations in the final analysis if possible.

==> Thank you for your kind words. As you noted, we will present the results of treatment details of pharmacopuncture and acupuncture, such as type and dosage of pharmacopuncture and used acupoints, and we will consider treatment variations when analyzing the results. Additionally, we will consult with statistical experts to explore how these variations can be investigated through statistical analyses.

However, as the trial is allowing the health providers to personalize the interventions and protocols (both for pharmacopuncture and acupuncture) to the participant's clinical status, I would recommend including the variations in the final analysis if possible.

==> Thank you for your kind words. As you noted, we will present the results of treatment details of pharmacopuncture and acupuncture, such as type and dosage of pharmacopuncture and used acupoints, and we will consider treatment variations when analyzing the results. Additionally, we will consult with statistical experts to explore how these variations can be investigated through statistical analyses.

Reviewer 3 Report

Dear Authors,

Thank you for submitting your study protocol paper "Pharmacopuncture effects on insomnia disorder: a multi-site, randomized, acupuncture-controlled, clinical study protocol"

This was an interesting study protocol, however I do have some substantial concerns which need to be addressed by a major revision before I can recommend publication.

Title

1. I would suggest rewording the second sentence of your title slightly "Pharmacopuncture effects on insomnia disorder: Protocol for a multi-site, randomized, acupuncture-controlled clinical trial"

Abstract

2. Line 1 "insomnia is a common medical condition" - Is insomnia a medical or psychological/psychiatric condition? I would dispute that it is a medical condition as there is typically no abnormal medical investigation findings in insomnia.

3. Abstract will require updating in response to other methodological concerns I will discuss in further detail later in this review (e.g., selection of primary outcome measure).

Introduction

4. Line 49 - 50, when referencing insomnia and medical conditions, please list some examples

5. The sentence lines 51 - 54 is inaccurate. The latest evidence, and the American Academy of Sleep Medicine guideline and position statement is that Psychological and Behaviour interventions should be used as the initial treatment (https://doi.org/10.5664/jcsm.27286) Reference 13 is a guideline specifically on the use of pharmacological management of insomnia, it will, of course, not address non-pharmacological strategies. Please revise this sentence. I see in the next sentence you talk about CBT-i, which is good, but sentence before discussing CBT-i needs to be amended or deleted.

6. Side effects of medication is mentioned on line 57-58, but not specified what these are. Please revise.

7. Sentence line 63-65 is not quite true, there are studies emerging about CBT-i and medical comorbidity, e.g., sleep apnoea (https://doi.org/10.1093/sleep/zsaa041) ,arthritis, and coronary artery disease (https://doi.org/10.1037/a0025577).

8. Lines 70 - 73, regarding the Korean guideline for insomnia and acupuncture, does this guideline also recommend the gold standard treatment (CBT-i)? If not, this should be appended to the sentence as a limitation of the guideline, because any credible guideline would recommend CBT-i.

9. Line 76 - what is meant by quick effect? probably better to specify what this refers to pharmacokinetics? subjective effect for the patient?

10. Sentence on line 82 - 84, sleep quality is very broad, how was it measured in that study?

Method

11. Line 98 - "RCT was conducted". Could the authors please confirm if this is 1) an error and should read "will be conducted" or 2) whether the study has already commenced or finished, and the authors are seeking to publish the protocol despite the advanced stage of the trial. If the trial has already commenced, the authors need to explain the current progress in the trial and why the protocol was not submitted for peer review before the trial began.

12. Authors need to justify the very specific number of 138 patients to be recruited. I presume this is based on a power analysis, and if so, that should be reported. EDIT: I see this is later discussed. I have concerns about the use of an unpublished masters' degree thesis from 8 years ago which has not been peer reviewed as the justification for the sample size. Please report a power analysis showing this sample size will be adequate.

13. I think a longer follow-up period would be more advantageous than 4 weeks. What would be more helpful is to see if any improvements were sustained several months later.

14. How will the 15 participants be selected for polysomnography and other biological measures? This needs to be specified in the protocol, including how the authors will statistically test to see if those 15 participants differed significantly from the rest of the group on key demographic and outcome measures. 

15. Figure 1 - please justify why the pharmacopuncture group will be twice the sample size of the active control acupuncture group. EDIT: I see this is discussed later on in the methods, so maybe when that 2:1 ratio is first mentioned, just direct the reader to the sample section.

16. Line 145 - who will diagnose insomnia? Which clinical profession? - please specify

17. line 146-147 - participants also need to be full informed of any risks and potential for adverse effects from the trial procedures - please amend

18. Line 180 - your exclusion criteria is already quite thorough, could you specify in a bit more detail about the discretion the investigator has for excluding a patient?

19. The blinding is difficult in a study like this. I'm pleased to see that the assessors will be blinded, even though the clinicians administering treatment cannot be blinded.

20. Line 198 to 206 and 2017 - 210 - I haven't got to your discussion section yet, but this variability in the administration between patients should be listed as a confound and limitation.

21. Line 238 - I think the selection of PSQI as the primary measure is wrong. Whilst insomnia will cause a higher PSQI score, it is not a specific measure of insomnia. I strongly suggest changing the primary outcome to an insomnia specific measure, and keeping PSQI as a secondary outcome.

22. line 288 Please specify more about how cost data will be calculated.

23. Because this is a protocol paper, I think more needs to be explained in the secondary outcomes about what those outcomes actually will be. E.g., actigraphy is mentioned but not what the outcome of that will be, such as reduced sleep latency, increased total sleep time etc. Perhaps a table in the secondary outcome section with specific outcomes.

24. Line 373 - I think some regression analyses could also be planned as well. It needs to be justified that any effects found due to the treatment withstand the variance accounted for by other covariates in the model.

25. I think more needs to be said about the assessors and their clinical qualifications, as well as those practitioners administering the treatment.

26. Okay, so now you've mentioned about the trial status. Could the authors please justify why this protocol is only being submitted for publication now? This complicates things because if the trial has been going for nearly a year it means the protocol would need to be varied to address some of my concerns (e.g., primary outcome measure, counselling patients on side effects prior to starting the study etc).

27.  The limitations section is far too brief and needs to be expanded, including addressing a number of the methodological concerns noted in my review.

28. First sentence of the conclusion, it would be good if you could cite any other studies showing the hypothesised "synergistic effect".

Author Response

Reviewer 3.

Dear Authors,

Thank you for submitting your study protocol paper "Pharmacopuncture effects on insomnia disorder: a multi-site, randomized, acupuncture-controlled, clinical study protocol"

This was an interesting study protocol, however I do have some substantial concerns which need to be addressed by a major revision before I can recommend publication.

==> We are grateful for your comments; they have greatly helped in improving this manuscript. We have done our best to address all your concerns and revised the manuscript to incorporate your suggestions as much as possible.

Title

  1. I would suggest rewording the second sentence of your title slightly "Pharmacopuncture effects on insomnia disorder: Protocol for a multi-site, randomized, acupuncture-controlled clinical trial“

==> As per your suggestion, we have reworded the title to “Pharmacopuncture effects on insomnia disorder: Protocol for a multi-site, randomized, acupuncture-controlled clinical trial.”

(Lines 2 & 3)

Abstract

  1. Line 1 "insomnia is a common medical condition" - Is insomnia a medical or psychological/psychiatric condition? I would dispute that it is a medical condition as there is typically no abnormal medical investigation findings in insomnia.

==> Thank you for highlighting this point. We have replaced the phrase “a common medical condition” with “a common health problem.”

Lines 23 & 24

“Insomnia is a common health problem that can lead to various diseases and negatively impact quality of life.”

  1. Abstract will require updating in response to other methodological concerns I will discuss in further detail later in this review (e.g., selection of primary outcome measure).

==> Thank you for your detailed advice. Unfortunately, as this study is in progress, it was difficult to majorly revise the methodological constituents, such as the primary outcome. However, we have done our best to address your methodological concerns and have noted all relevant limitations in the discussion section.

Introduction

  1. Line 49 - 50, when referencing insomnia and medical conditions, please list some examples

==> As suggested, we have added some examples of medical conditions associated with insomnia.

Lines 47-51

“Insomnia is linked to reduced productivity, poor job or academic performance, an increased risk of workplace or traffic accidents, and an increased vulnerability to a variety of medical conditions such as psychiatric disorders, cognitive impairment, cardiovascular diseases, and metabolic diseases [7-11].”

  1. The sentence lines 51 - 54 is inaccurate. The latest evidence, and the American Academy of Sleep Medicine guideline and position statement is that Psychological and Behaviour interventions should be used as the initial treatment (https://doi.org/10.5664/jcsm.27286) Reference 13 is a guideline specifically on the use of pharmacological management of insomnia, it will, of course, not address non-pharmacological strategies. Please revise this sentence. I see in the next sentence you talk about CBT-i, which is good, but sentence before discussing CBT-i needs to be amended or deleted.

==> We have deleted the inaccurate portions of the text as suggested.

Lines 62-65

“Pharmacotherapy, in particular, benzodiazepines are effective for the short-term (3–4 weeks) treatment of insomnia [20]. However, clinical guidelines state that short-term hypnotics administration should be supplemented with behavior and cognitive therapies when possible [4].”

  1. Side effects of medication is mentioned on line 57-58, but not specified what these are. Please revise.

==> As you suggested, we have added a list of side effects caused by hypnotics.

Lines 65-68

“Moreover, long-term use of benzodiazepine increases the risk of abuse, tolerance, dependence, and associated medical complications such as falls, fractures, and impaired attention, psychomotor function, and cognitive function especially in older adults [21-24]."

  1. Sentence line 63-65 is not quite true, there are studies emerging about CBT-i and medical comorbidity, e.g., sleep apnoea (https://doi.org/10.1093/sleep/zsaa041)

,arthritis, and coronary artery disease (https://doi.org/10.1037/a0025577).

==> We have deleted the inaccurate portions of text as per your comment.

Lines 70-77

“CBT-I is not only effective in improving insomnia [25, 26] but also comorbid insomnia with sleep apnea, arthritis, and coronary artery disorder [27,28]. However, some patients fail to achieve any benefit from this treatment [29], and there are some challenges in implementing CBT-I, including the lack of trained therapists, low patient motivation, and high burden of cost and time [29]. Recently, internet-based CBT-I is shown to be as effective as in-person practitioner-administered CBT-I, and there are debates on the extent of therapist’s involvement in this treatment [30].”

  1. Lines 70 - 73, regarding the Korean guideline for insomnia and acupuncture, does this guideline also recommend the gold standard treatment (CBT-i)? If not, this should be appended to the sentence as a limitation of the guideline, because any credible guideline would recommend CBT-i.

==> Accordingly, we have revised the limitation section of the manuscript as follows:

Lines 83-88

“The Clinical Practice Guideline of Korean Medicine for Insomnia Disorder states that acupuncture, as monotherapy and in combination with sleeping pills, should be considered in treating insomnia. The Clinical Practice Guideline of Korean Medicine for Insomnia Disorder was the first of its kind in South Korea to recommend traditional Korean medical interventions for treating insomnia. However, it was limited in that it did not recommend CBT-I as an initial treatment [34]."

  1. Line 76 - what is meant by quick effect? probably better to specify what this refers to pharmacokinetics? subjective effect for the patient?

==> As suggested, we have added a detailed explanation for the term “quick effect” in the manuscript.

Line 89-94

“Pharmacopuncture is a new type of acupuncture therapy that involves injecting herbal medicine extracts into acupoints, thus producing a combination of mechanical and chemical effects. Its benefits include a relatively quick effect and ease of administered dosage control. Additionally, since the herbal medicine is administered and absorbed directly into the acupoints without direct involvement of the gastrointestinal tract, this treatment can be extended to patients who cannot consume herbal medicine orally [35].”

  1. Sentence on line 82 - 84, sleep quality is very broad, how was it measured in that study?

==> As suggested, we have added a description for how sleep quality was measured.

Line 99-101

“A systematic review reported that the decrease in PSQI score after pharmacopuncture therapy was significantly greater than that after pharmacotherapy or acupuncture [37].”

Method

  1. Line 98 - "RCT was conducted". Could the authors please confirm if this is 1) an error and should read "will be conducted" or 2) whether the study has already commenced or finished, and the authors are seeking to publish the protocol despite the advanced stage of the trial. If the trial has already commenced, the authors need to explain the current progress in the trial and why the protocol was not submitted for peer review before the trial began.

==> Kindly note that this trial is ongoing, and recruitment is expected to end by late 2023. We agree that the protocol should have been peer reviewed before beginning recruitment. However, we faced several barriers in the process. Firstly, some outcome measurements needed to be revised and we needed to re-submit the protocol for IRB’s approval of the changes; this delayed our submission. Therefore, even though the final version of the protocol was updated and ready, waiting for IRB approval somewhat delayed the submission of our manuscript. Secondly, before submitting to the International Journal of Environmental Research and Public Health, we had submitted our protocol to several other journals that rejected the manuscript, this inevitably delayed the submission.

We acknowledge this delay as a limitation of our study, and we have incorporated your concerns in the limitation section of the manuscript. We will consider your methodological concerns in the data analysis phase of the study and note them in the study discussion, when completed. Further, we will refer to your comments and advice in our future research design.

  1. Authors need to justify the very specific number of 138 patients to be recruited. I presume this is based on a power analysis, and if so, that should be reported. EDIT: I see this is later discussed. I have concerns about the use of an unpublished masters' degree thesis from 8 years ago which has not been peer reviewed as the justification for the sample size. Please report a power analysis showing this sample size will be adequate.

==> To calculate the adequate sample size for our study, we searched similar RCTs examining the effectiveness of pharmacopuncture on insomnia disorder but could not find comparable published studies that were peer reviewed. Therefore, we calculated the sample size of this trial based on a study with similar design and the same primary outcome.

  1. I think a longer follow-up period would be more advantageous than 4 weeks. What would be more helpful is to see if any improvements were sustained several months later.

==> Because this study is already ongoing, it is difficult to revise the follow-up period. We acknowledge that the short follow-up period is a limitation of this study and we will include it under the limitations in the discussion section. Additionally, in further research on this topic, we will plan adequate follow-up period such as several months.

Lines 493-497

“Third, the 4-week follow-up period in this trial is not sufficient to evaluate the persisting therapeutic effect of pharmcopuncture. Future studies with an adequate follow-up period of several months are needed to investigate the long-term therapeutic effects of pharmcopuncture or acupuncture.”

  1. How will the 15 participants be selected for polysomnography and other biological measures? This needs to be specified in the protocol, including how the authors will statistically test to see if those 15 participants differed significantly from the rest of the group on key demographic and outcome measures.

==> We will select consecutive 15 participants in each group enrolled at the Pusan National University Korean Medicine Hospital after random randomly assignment. We will record the preliminary (at the first visit) and post-intervention (at visit #11) measurements and conduct paired t-tests and two-sample t-tests to examine differences between baseline and post-intervention findings.

Line 146-150

“Additionally, 15 consecutive participants from each group enrolled at the Pusan National University Korean Medicine Hospital after random randomly assignment will undergo polysomnography, melatonin and cortisol saliva studies on the day before the initiation of treatment (visit 1) and at the post-intervention evaluation (visit 11).”

Unfortunately, investigating whether there is a significant difference in demographic and outcomes variables between the 15 participants who will be selected for polysomnography and the rest of the cohort is beyond the scope of the current trial. Since polysomnography and saliva characteristics are additional explanatory outcomes and examining the effects of pharmacopuncture and acupuncture on sleep architecture using these variables is not our main research objective, we do not have a sophisticated analytical plan ready for addressing this research question. However, if the demographic and other outcomes variables significantly differ between participants who undergo polysomnography and the rest of the cohort, then we will perform further comparisons and describe the results in detail.

  1. Figure 1 - please justify why the pharmacopuncture group will be twice the sample size of the active control acupuncture group. EDIT: I see this is discussed later on in the methods, so maybe when that 2:1 ratio is first mentioned, just direct the reader to the sample section.

==> As suggested, we have described the sample size determination in the Methods section right after the section on participant recruitment and study schedule.

Please refer to lines 164-176; Section 2.3 Sample size.

  1. Line 145 - who will diagnose insomnia? Which clinical profession? - please specify

==> We have added details regarding the clinical proficiency and credentials of the investigators.

Lines 139-140

“The investigators will then diagnose the insomnia disorder according to DSM-5 criteria.”

  1. line 146-147 - participants also need to be full informed of any risks and potential for adverse effects from the trial procedures - please amend

==> Thank you for this advice. Of course, any risks and potential for adverse effects from the trial procedures are included in the informed consent forms and are fully explained to the participants when obtaining their signed consent. We have further amended the forms as per your advice.

Lines 134-137, 183-185

“The investigators, who are qualified neuropsychiatrists of Korean medicine, will explain the purpose, treatments, procedures of the clinical trial, and any risks and potential for adverse effects that may occur during this trial procedure.”

“(4) Voluntary participation and signing of an informed consent form after adequate explanation of the purpose and procedure of this clinical trial and any risks and potential for adverse effects from this trial procedure.”

  1. Line 180 - your exclusion criteria is already quite thorough, could you specify in a bit more detail about the discretion the investigator has for excluding a patient?

==> In principle, the participants are selected based on the inclusion/exclusion criteria described in the manuscript. Potential participants can be excluded if the investigators—who are neuropsychiatrists of Korean medicine—determine that preexisting medical conditions of the potential participants is inappropriate for this study. If necessary, a discussion between investigators will be undertaken to reach a consensus regarding inclusion/exclusion of special cases.

  1. The blinding is difficult in a study like this. I'm pleased to see that the assessors will be blinded, even though the clinicians administering treatment cannot be blinded.

==> Thank you for your positive comment. We will do our best to keep assessor blinded to the random assignment and interventions of participants.

  1. Line 198 to 206 and 2017 - 210 - I haven't got to your discussion section yet, but this variability in the administration between patients should be listed as a confound and limitation.

==> Personalization of treatments based on the patient's clinical condition must be considered when analyzing the results. In the limitations section, we acknowledge that we will not be able to examine the effect of improving insomnia of a specific type of phamarcopuncture or acupuncture. Once the trial is complete, we will be able to articulate and present details of the treatment performed during the trial.

Lines 483-490

“First, it is not possible to examine which type or acupoint in pharmacopuncture has the most significant effect on insomnia improvement, as we will allow for different types or methods of pharmacopuncture according to the practitioner’s clinical decision, depending on the patient’s condition. However, this policy can better reflect real-world Korean medical clinical practice, in which individualized treatment is performed. A retrospective analysis of medical records may provide complementary information regarding treatment details. We will detail the effects of these variations in the analysis and report the effectiveness of pharmacopuncture for insomnia.”

  1. Line 238 - I think the selection of PSQI as the primary measure is wrong. Whilst insomnia will cause a higher PSQI score, it is not a specific measure of insomnia. I strongly suggest changing the primary outcome to an insomnia specific measure, and keeping PSQI as a secondary outcome.

==> Because this study is already in progress, it is difficult to change the primary outcome. In our research experience, when we considered ISI as the primary outcome, we could only measure the subjective perception of the severity of insomnia, and it was challenging to assess important determinants of insomnia improvement such as sleep onset latency, total sleep time, and sleep efficiency. Although PSQI is also a self-reporting questionnaire, it can assess important sleep-related parameters economically and conveniently. Furthermore, PSQI has been used as a primary outcome in many previous RCTs on insomnia disorder. Thus, we have decided to retain PSQI score as the primary outcome variable in our study. Besides PSQI, sleep-related measurements like ISI, sleep diary, and actigraphy are used in our study to complement the interpretation of the effectiveness and safety of pharmacopuncture and acupuncture for insomnia disorder.

Lines 506-513

“Finally, the insomnia-specific objective assessment tools will not be used as a primary outcome in this study. Although PSQI—the primary outcome variable in this trial—is not an insomnia-specific measurement, it allows convenient evaluation of sleep onset latency, total sleep time, and sleep efficiency, which are important parameters for assessing sleep improvement. In addition, we use ISI to examine the subjective perception of insomnia severity and actigraphy to objectively evaluate sleep. Thus, we expect to obtain subjective and objective data on the sleep improvement effects of pharmacopuncture and acupuncture.”

  1. line 288 Please specify more about how cost data will be calculated.

==> We have added a detailed description of the cost data, as per your suggestion.

Lines 336-347

In this study, direct medical and non-medical costs and costs of productivity loss will be considered. Direct medical costs include expenses associated with pharmacopuncture, prescribed hypnotics, first or revisit diagnoses, and examination costs. Direct non-medical cost consists of the round-trip expenses for participating in clinical trials. The productivity loss costs will be calculated by multiplying the lost productivity scores by the hourly wage. The Work Productivity and Activity Impairment (WPAI) questionnaire will be used to calculate the lost productivity scores [54].

The result of the cost-effectiveness analysis will be expressed as the incremental cost-effectiveness ratio (ICER). This economic evaluation will include all types of costs and consider the quality-adjusted life-years (QALY) for effectiveness. QALY will be calculated using the EQ-5D-5L Gross domestic product (GDP) per capita will be used as the threshold value for willingness to pay.

  1. Because this is a protocol paper, I think more needs to be explained in the secondary outcomes about what those outcomes actually will be. E.g., actigraphy is mentioned but not what the outcome of that will be, such as reduced sleep latency, increased total sleep time etc. Perhaps a table in the secondary outcome section with specific outcomes.

==> As suggested, we have added more details on what sleep-related data will be obtained from the participants using actigraphy.

Lines 321-325

“We will calculate the total time in bed from the sleeping time and waking time recorded in the sleep diary. The sleep diary assesses the sleep onset latency and total sleep time. We will also calculate sleep efficiency from the items entered in the sleep diary. Additionally, we will analyze objective sleep-related data such as sleep onset latency, total sleep time, and sleep efficiency based on actigraphy findings.”

  1. Line 373 - I think some regression analyses could also be planned as well. It needs to be justified that any effects found due to the treatment withstand the variance accounted for by other covariates in the model.

==> We have revised the manuscript as per your suggestion. If there any significant between-group differences are observed in baseline variables or demographic characteristics, regression analysis with covariates (analysis of covariance) will be conducted.

Lines 421-423

“If there were significant between-group differences in baseline variables or demographic characteristics, regression analysis with covariates (analysis of covariance) will be conducted.”

  1. I think more needs to be said about the assessors and their clinical qualifications, as well as those practitioners administering the treatment.

==> We have added the relevant information in the manuscript.

Lines 231-234

“The assessors participating in this study are residents at the Department of Neuropsychiatry of Korean Medicine and clinical research coordinators who have completed research ethics education and standard operating procedure training related to this trial before trial initiation.”

Line 262-265

"Both pharmacopuncture and acupuncture will be performed by Korean medicine doctors with more than 2 years of experience in acupuncture practice. All practitioners will have completed the standard operating procedure training related to pharmachopuncture and acupuncture administration before this trial’s initiation."

  1. Okay, so now you've mentioned about the trial status. Could the authors please justify why this protocol is only being submitted for publication now? This complicates things because if the trial has been going for nearly a year it means the protocol would need to be varied to address some of my concerns (e.g., primary outcome measure, counselling patients on side effects prior to starting the study etc).

==> As described in our response to question #11 and in the main text of the manuscript, the status of this study is ongoing. Kindly also consider the delay that occurred in the preparation of the manuscript owing to the emergent need for additional approval from the IRB for changes made in the protocol after study initiation and that this manuscript was rejected by other journals before submission to IJERPH.

Among the methodological concerns you have stated, we have addressed question #21 regarding the primary outcome selection and described it as a limitation in the discussion section. In addition, we have addressed question #17 regarding the counseling of patients on side effects prior to starting the study. All investigators explain the purpose, procedure, and risks of intervention and the potential of advertise effects during trial to each potential participant before obtaining their signed informed consent. The IRB-approved consent form of this study also clearly state the risks and potential adverse effects associated with participating in this trial procedure. All participants are first made fully aware of the purpose, procedures, risks of intervention, and potential adverse effects of the trial, and only then are they asked to sign the informed consent form.

  1. The limitations section is far too brief and needs to be expanded, including addressing a number of the methodological concerns noted in my review.

==> We agree with the reviewer and have rewritten this section to include all relevant limitations of our protocol.

Lines 483-513

“However, this study has some limitations. First, it is not possible to examine which type or acupoint in pharmacopuncture has the most significant effect on insomnia improvement, as we will allow for different types or methods of pharmacopuncture according to the practitioner’s clinical decision, depending on the patient’s condition. However, this policy can better reflect real-world Korean medical clinical practice, in which individualized treatment is performed. A retrospective analysis of medical records may provide complementary information regarding treatment details. We will detail the effects of these variations in the analysis and report the effectiveness of pharmacopuncture for insomnia. Second, we will only recruit patients with insomnia disorder; therefore, the generalizability of the results to insomnia patients with other comorbidities, including depression, anxiety disorders, alcohol dependence, and other disorders, might be limited. Third, the 4-week follow-up period in this trial is not sufficient to evaluate the persisting therapeutic effect of pharmcopuncture. Future studies with an adequate follow-up period of several months are needed to investigate the long-term therapeutic effects of pharmcopuncture or acupuncture. Fourth, both participants and practitioners cannot be blinded to treatments because a sham acupuncture group could not be designed in this study. Therefore, we need to consider factors affecting treatment, such as participants' expectations from pharmacopunture treatment, when interpreting the results of the study. Since the purpose of this trial is to investigate the effectiveness and safety of pharmacopuncture of uninsured treatment and acupuncture covered by NHI, this trial will not include a comparison between the intervention group and a third group treated with placebo or standard treatment as per Western medicine such as CBT-I. Further 3-arm trials including a placebo group to investigate the efficacy of pharmcopuncture or CBT-I group to compare the effectiveness of that to standard treatment are needed. Finally, the insomnia-specific objective assessment tools will not be used as a primary outcome in this study. Although PSQI—the primary outcome variable in this trial—is not an insomnia-specific measurement, it allows convenient evaluation of sleep onset latency, total sleep time, and sleep efficiency, which are important parameters for assessing sleep improvement. In addition, we use ISI to examine the subjective perception of insomnia severity and actigraphy to objectively evaluate sleep. Thus, we expect to obtain subjective and objective data on the sleep improvement effects of pharmacopuncture and acupuncture.”

  1. First sentence of the conclusion, it would be good if you could cite any other studies showing the hypothesised "synergistic effect".

==> We have revised this statement to use a more appropriate expression.

Lines 515-516

“Pharmacopuncture is expected to be effective in improving insomnia due to the combined effects of acupuncture and herbal medicine.”

Round 2

Reviewer 1 Report

Authors improve their discussion and aligned their results with the main manuscript body. I agree with the current version and nothing more to add.

Author Response

Authors improve their discussion and aligned their results with the main manuscript body. I agree with the current version and nothing more to add.

-- Thank you very much for your kindness.